# On the treatment effect heterogeneity of antidepressants in major depression: A Bayesian meta-analysis and simulation study

**Constantin Volkmann**[1]ම*, **Alexander Volkmann**[2]ම, **Christian A. Müller**[1]

**1** Department of Psychiatry and Psychotherapy, Charité—Universitätsmedizin Berlin, Campus Charité Mitte, Berlin, Germany, **2** Independent Researcher, Berlin, Germany

ම These authors contributed equally to this work.

\* constantin.volkmann@charite.de

**Data Availability Statement:** The data underlying the results presented in the study are available from: https://data.mendeley.com/datasets/83rthbp8ys/2

## Abstract

### Background

The average treatment effect of antidepressants in major depression was found to be about 2 points on the 17-item Hamilton Depression Rating Scale, which lies below clinical relevance. Here, we searched for evidence of a relevant treatment effect heterogeneity that could justify the usage of antidepressants despite their low average treatment effect.

### Methods

Bayesian meta-analysis of 169 randomized, controlled trials including 58,687 patients. We considered the effect sizes log variability ratio (lnVR) and log coefficient of variation ratio (lnCVR) to analyze the difference in variability of active and placebo response. We used Bayesian random-effects meta-analyses (REMA) for lnVR and lnCVR and fitted a random-effects meta-regression (REMR) model to estimate the treatment effect variability between antidepressants and placebo.

### Results

The variability ratio was found to be very close to 1 in the best fitting models (REMR: 95% highest density interval (HDI) [0.98, 1.02], REMA: 95% HDI [1.00, 1.02]). The between-study standard deviation $\tau$ under the REMA with respect to lnVR was found to be low (95% HDI [0.00, 0.02]). Simulations showed that a large treatment effect heterogeneity is only compatible with the data if a strong correlation between placebo response and individual treatment effect is assumed.

### Conclusions

The published data from RCTs on antidepressants for the treatment of major depression is compatible with a near-constant treatment effect. Although it is impossible to rule out a substantial treatment effect heterogeneity, its existence seems rather unlikely. Since the

**Funding:** We acknowledge support from the German Research Foundation (DFG) and the Open Access Publication Fund of Charité – Universitätsmedizin Berlin.

**Competing interests:** CAM received consulting fees from Silence Therapeutics, outside the submitted work. The other authors declared no competing interest. This does not alter our adherence to PLOS ONE policies on sharing data and materials.

average treatment effect of antidepressants falls short of clinical relevance, the current prescribing practice should be re-evaluated.

## Introduction

Major depressive disorder is one of the most frequent psychiatric conditions and poses a major burden for individuals and society; it affects more than 300 million people worldwide and is ranked as the single largest contributor to disability [1]. The first-line treatment usually consists of psychotherapy and/or pharmacotherapy with antidepressant drugs [2, 3]. Within the last decades, the number of prescriptions of antidepressants has continuously increased in several regions of the world. A recent meta-analysis by Cipriani et al. comprising 522 randomized, controlled trials (RCTs) of 21 antidepressants in 116,477 participants reported that all antidepressants were more effective than placebo in reducing depressive symptoms [4]. However, the so-called average treatment effect, which measures the difference in mean outcomes between active and control group, was only about 2 points on the 17-item Hamilton Depression Rating Scale (HAMD-17) [5] in this dataset [6]. A reduction of 2 points on the HAMD scale is probably not detectable by the treating physician and of questionable clinical relevance. According to Leucht et al. [7], a reduction of up to 3 points on the HAMD corresponds to "no change" in the Clinical Global Impressions—Improvement Scale (CGI-I) and the assumed threshold of clinical significance is between 3 and 7 points [8, 9].

### Treatment effect heterogeneity

Despite the small average treatment effect of antidepressants, it is commonly assumed that subpopulations of patients exist that have a clinically relevant benefit. However, a clinically relevant treatment effect heterogeneity, albeit widely believed and intuitively plausible, has not conclusively been shown to exist yet. The observed change in clinical symptomatology (the change or response score), which is highly variable, is the sum of different effects, such as placebo effect, regression to the mean, spontaneous remission and treatment effect [10, 11]. Hence, the true effect of an intervention cannot be observed in a single patient, as it would require the observation (and subtraction of) both potential outcomes–with and without the intervention. This Fig 1 illustrates the relationship between observed change, individual treatment effect and treatment effect heterogeneity.

### The variability ratio

In order to assess treatment effect heterogeneity from the data of parallel group trials, the comparison of outcome variances between the active and the control condition has been proposed [10]. Two different effect size statistics have been suggested to estimate the difference in variability between two groups [12]. The variability ratio (VR) is the quotient of the variances of the active and placebo groups, while the coefficient of variance ratio (CVR) corrects for differences in means between the groups. Here, an increase or a decrease in variance in the active group might be a signal of a varying individual treatment effect.

Following a recent publication by Winkelbeiner et al. [13], analyzing differences in variances in 52 randomized, placebo-controlled antipsychotic drug trials, this work aimed at estimating the degree of treatment effect heterogeneity of antidepressants using the open dataset of the largest meta-analysis of the efficacy of antidepressants in major depressive disorder. Three other research groups [14–16] have addressed the same question of treatment effect heterogeneity of antidepressants using the same dataset.

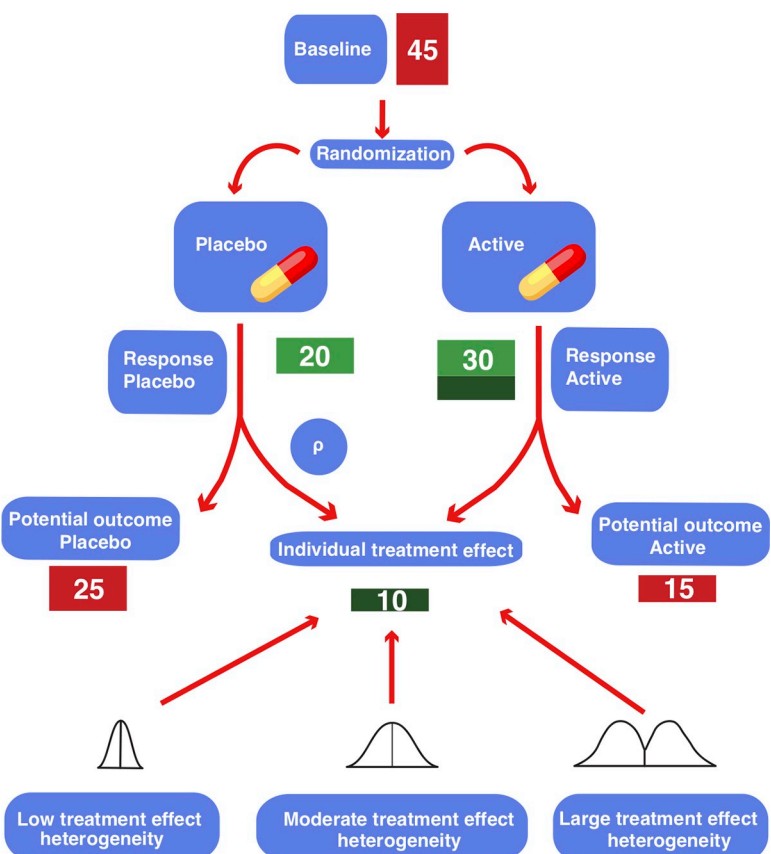

**Fig 1. Visualization of potential outcomes and treatment effect for a patient in an antidepressant trial.** The patient is randomized to either the placebo or the active arm, corresponding to two hypothetical "potential outcomes" (Red scores). Only one outcome can be observed, as a patient cannot receive both interventions simultaneously. The difference between the two outcomes is the "individual treatment effect" of the intervention (Dark green score). The individual treatment effect is unobservable and can be imaged to be drawn from hypothetical distributions of the treatment effect. The variance of this distribution corresponds to the treatment effect heterogeneity. The factor ρ is the correlation between the placebo response and the individual treatment effect. All numbers signify depression severity on the HAMD-17 scale.

However, all previous analyses did not sufficiently take into account the linear association between the logarithmic mean and logarithmic standard deviation (SD) of the underlying measuring scale when estimating the direct effect of antidepressants on outcome variability. The VR and the CVR effect sizes naively assume a slope coefficient of 0 and 1 on this association, respectively, and may thus yield biased results if the slope coefficient differs from the assumed one. Crucially, using the VR in this dataset suggests no evidence for treatment effect heterogeneity [14, 15], while using the CVR leads to the conclusion of a variable antidepressant treatment effect [16].

We reanalyzed the Cipriani dataset by implementing a Bayesian meta-regression that models the degree of linear association between lnMean and lnSD directly from the data and thus yields more accurate results. We illustrated the exact relationship between VR and CVR and their respective limitations and disentangled direct and indirect effect of antidepressants on outcome variability. We showed that the CVR is a suboptimal measure, as it makes strong assumptions that are not warranted in this dataset. Furthermore, we investigated the compatibility of the results with different degrees of treatment effect heterogeneity using simulations

experiments. Using an analytically derived formula, we provided an upper bound on the treatment effect heterogeneity based on the VR estimate.

## Methods

### Statement of ethics

The authors have no ethical conflicts to disclose.

### Data acquisition

We obtained the dataset of the meta-analysis by Cipriani et al. [4] from the Mendeley database (https://data.mendeley.com/datasets/83rthbp8ys/2). This study included 522 RCTs comparing 21 antidepressants with placebo or another active antidepressant as oral monotherapy for the acute treatment of adults ($\geq$18 years old and of both sexes) with a primary diagnosis of major depressive disorder according to standard operationalized diagnostic criteria (Feighner criteria, Research Diagnostic Criteria, DSM-III, DSM-III-R, DSM-IV, DSM-5, and ICD-10). For further details on the inclusion criteria and study characteristics, see the original study. We did not attempt to obtain individual patient data but relied on summary data from the openly accessible dataset.

### Data extraction and processing

Of the total of 522 studies we kept the 304 that included a placebo arm. We excluded all studies for which the reported endpoint did not represent the change from baseline, leaving us with a total of 169 studies including 58,687 patients for the analysis (see PRISMA flow diagram, Fig 2). We extracted both the mean and the standard deviation of pre- and post-treatment outcome difference scores (the "response"). The studies included in the data set comprised 8 different depression scales, namely HAMD-17, HAMD-21, HAMD-24, HAMD unspecified, HAMD-29, HAMD-31, Montgomery–Åsberg Depression Rating Scale (MADRS) [17] and IDS-IVR-30 [18]. Studies with different treatment arms were aggregated according to the recommendation of the Cochrane Collaboration. In this manuscript, we define response as pre-post-difference of a given outcome scale.

### Investigation of treatment effect heterogeneity

In the case of a constant treatment effect, the variances of the active and the placebo group are expected to be equal. Therefore, a large difference in variances in response to antidepressants and placebo may be regarded as evidence of treatment effect heterogeneity.

Two different effect size statistics have been suggested by Nakagawa et al. [12] to estimate the difference in variability of active and placebo response:

1. The *log variability ratio*

$$lnVR = \ln\left(\frac{SD_a}{SD_p}\right) + \frac{1}{2(n_a - 1)} - \frac{1}{2(n_p - 1)}, \text{ where :}$$

$$SD = standard\ deviation, a = active, p = placebo$$

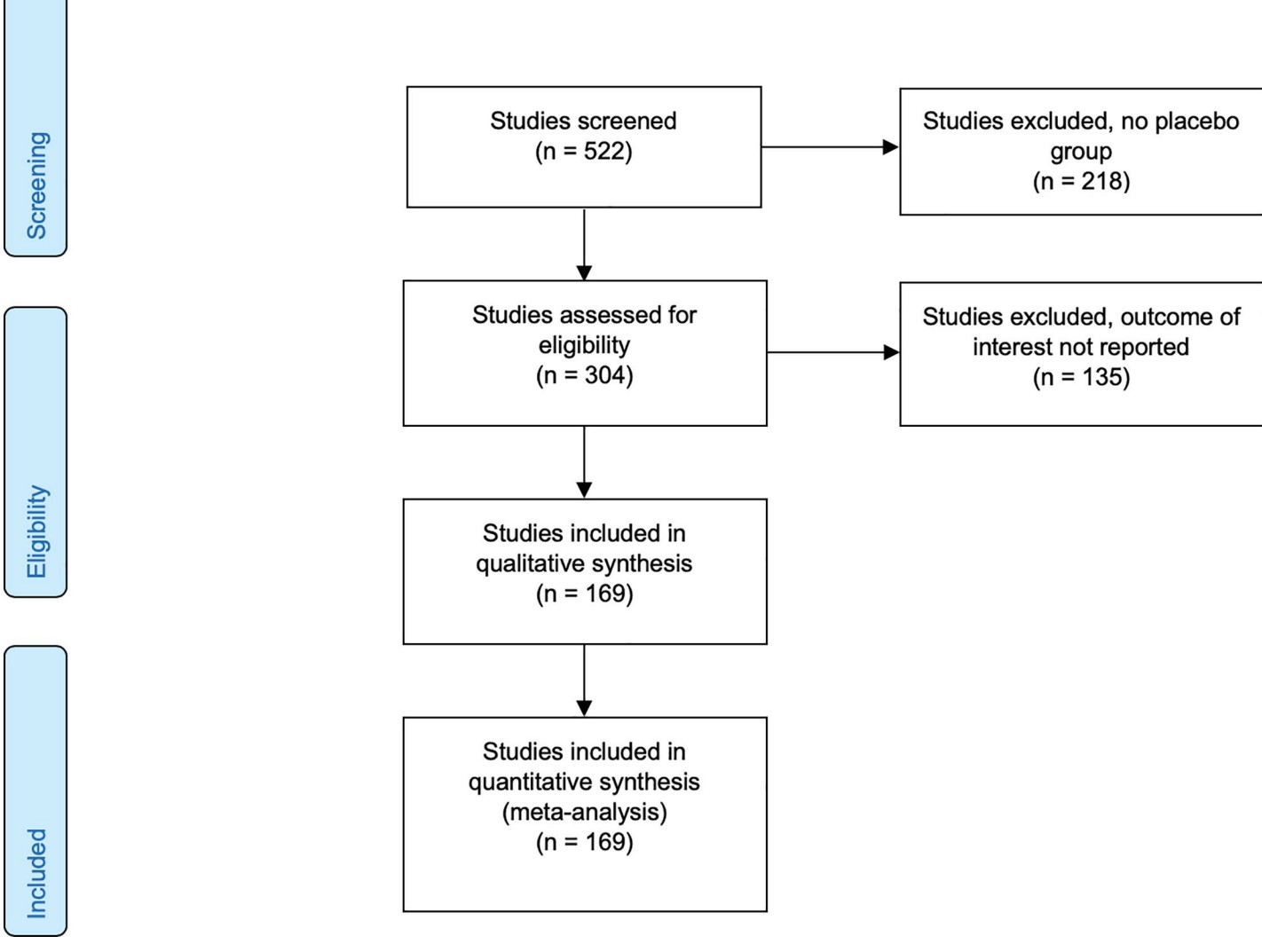

**Fig 2. PRISMA 2009 flow diagram.**

2. The *log coefficient of variation ratio*

$$lnCVR = \ln\left(\frac{CV_a}{CV_p}\right) + \frac{1}{2(n_a - 1)} - \frac{1}{2(n_p - 1)}, \text{ where}: \ CV = \frac{SD}{mean}$$

The VR measures the total effect (direct and indirect) of antidepressants on outcome variability. If the mean and standard deviation of the outcome scale are associated, the active treatment would have an indirect effect on the outcome variability mediated by its effect on the outcome mean. In order to estimate only the direct effect of the treatment on outcome variability, it is therefore necessary to control for the mean ~ SD relationship (Fig 3).

The lnCVR controls for such an association but implicitly assumes a linear relationship between lnMean and lnSD with slope coefficient of 1. For clarification, consider a linear

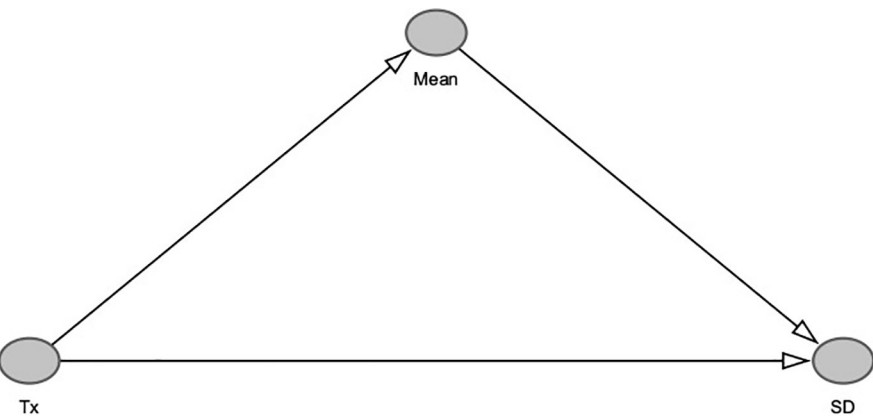

**Fig 3. Causal graph depicting relationship between treatment, mean and SD.** The treatment Tx has a causal effect of around 2 Hamilton points on the mean of the study population. If we control for the mean ~ SD relationship when estimating the variability between treatment and control, we only measure the direct effect of treatment on variability, while no such control (as with the VR effect size) measures the total effect of the treatment on outcome variability.

relationship between lnMean and lnSD across different studies with slope coefficient 1:

$$lnSD = const + lnMean$$

Then it follows that CV is constant across studies:

$$lnCV = ln\left(\frac{SD}{mean}\right) = const$$

Therefore, the lnCVR can be represented by a linear function between lnVR and lnRR with slope coefficient 1:

$$lnCVR = const_a - const_p = \alpha$$

$$lnVR - lnRR = \alpha$$

$$lnVR = \alpha + 1 * lnRR$$

Conversely, the lnVR can be represented by a linear function between lnVR and lnRR with slope coefficient 0:

$$lnVR = \alpha + 0 * lnRR$$

Where lnRR is defined as:

3. The *log of the response ratio*

$$lnRR = ln\left(\frac{mean_a}{mean_p}\right), \text{ where mean denotes the mean of the response variable}$$

The lnCVR may therefore give biased results when estimating the direct effect, if the slope coefficient of this linear association differs from 1. Conversely, the lnVR effect size may yield biased results, if one is interested in the direct effect only and an lnMean ~ lnSD relationship exists. This is a relevant limitation of both effect sizes that has not been considered by previous analyses.

## Linear association between lnMean and lnSD

The CVR effect size implicitly assumes a linear association between the logarithm of the mean response scores (lnMean) and the logarithm of their standard deviation (lnSD) with a slope coefficient of 1. In order to test this assumption, we investigated this association by using a varying intercept model, where the intercepts were allowed to vary between studies with different depression scales. Moreover, testing for a linear association between lnMean and lnSD helps assessing the validity of the exchangeability assumption of random-effects meta-analyses. If there is a strong influence of lnMean on lnSD, the lnVR meta-analysis may yield biased results [19].

## Statistical analysis

All statistical analyses were carried out in the programming language Python (version 3.7) and the probabilistic programming language Stan (with pystan version 2.19.1.1 as a Python interface). The python code is available at: https://github.com/volkale/adv.

We used a Bayesian approach to fit all our models using weakly informative priors. Firstly, we used a Bayesian random-effects meta-analyses (REMA) for the two effect statistics lnVR and lnCVR. Secondly, we used a Bayesian random-effects meta-regression (REMR) to fit the lnVR effect statistic with the natural logarithm of the response ratio (lnRR) as a regressor. An additional complexity in our analysis, as compared to recent analyses [13, 20], came from the fact that our data set contained several different depression scales (several versions of the HAMD and the MADRS, see S1 Fig). For our analysis we made the assumption that these different scales are (locally) linearly transformable into each other. This assumption is well supported by the literature [21]. Fortunately, the lnVR and lnCVR effect statistics are invariant under linear transformations of the outcome scale.

## Random-effects meta-analysis (REMA)

We applied a Bayesian random-effects meta-analysis in order to estimate the effect sizes lnVR and lnCVR. For the REMA, the following model was applied, where μ equals the underlying mean of the effect size. Finally, $\tau^2$ represents the between-study-variance.

$$ES_i \sim N(\alpha_i, SD_i^2)$$

$$\alpha_i \sim N(\mu, \tau^2)$$

We specified the following weakly-informative hyper-priors:

$$\mu \sim Cauchy(0, 1)$$

$$\tau \sim HalfCauchy(0, 1)$$

## Random-effects meta-regression (REMR)

This approach is a "contrast-based" version of the "arm-based" meta-analysis in Nakagawa et al. [12] which models the log of the standard deviation of the outcome directly in a multi-level meta-regression. The advantage of this model over the lnVR and lnCVR meta-analyses is that we are not forced to make rigid assumptions about the lnMean ~ lnSD association, as the strength of this relationship is estimated directly from the data. The REMR models this relationship as follows:

$$lnVR = \alpha + \beta * lnRR$$

Note that the lnVR corresponds to a β coefficient set to 0, whereas the lnCVR corresponds to a β coefficient set to 1. In the REMR, the slope β can take any real value and is estimated

from the data. The contrast-based approach takes care of the issue of the occurrence of different depression scales, as it considers relative differences between active and placebo arms within each study. Since the depression scales are linearly transformable into each other, those relative differences do not depend on the depression scale.

We used the following model, where $lnVR_i$ is the measured lnVR of study i and $lnv_i$ is the true value of $lnVR_i$. $\alpha_i$ is the study-specific direct effect, whereas $\mu$ denotes the meta-analytic direct effect. $\beta$ is the slope coefficient of $lnr_i$ (the true value of $lnRR_i$), $\beta*lnr_i$ thus equals the study-specific indirect effect.

$$lnVR_i \sim N(lnv_i, s^2_{lnVR_i})$$

$$lnv_i = \alpha_i + \beta * lnr_i$$

$$\alpha_i \sim N(\mu, \tau^2)$$

$$lnr_i \sim N(0, 2.5^2)$$

$$lnRR_i \sim N(lnr_i, s^2_{lnRR_i})$$

We specified the following weakly-informative hyper-priors:

$$\mu \sim Cauchy(0, 1)$$

$$\beta \sim Cauchy(0, 1)$$

$$\tau \sim HalfCauchy(0, 1)$$

## Random-effects vs. fixed-effect model

When investigating the heterogeneity of a treatment effect, the choice between a random-effects and a fixed-effect approach has relevant consequences for the interpretation of the results. The random-effects model assumes that the treatment effect varies between populations and studies and hence estimates the average intervention effect [22]. It is a matter of interpretation, if we consider the between-study-variance to be a results of sampling error or a true variation of the effect. In the latter case, the between-study variance $\tau^2$ and not only the mean variability may be considered in order to estimate treatment effect heterogeneity.

## Choice of priors

Since the number of studies in this meta-analysis was very large relative to the number of model parameters, we chose regularizing, minimally informative priors. In order to test the sensitivity of the results to the choice of prior, we repeated the main analysis (REMR) using an enthusiastic prior as suggested by Spiegelhalter et al. [23]. Specifically, we used a normal prior centered at $e^\mu = 2$ ($e^\mu$ being the meta-analytic direct effect on the non-log scale) with a variance such that the prior probability of $e^\mu$ being smaller than 1 was less than 5%. This can roughly be interpreted as expecting a VR of larger than 1 with 95% certainty and a mean estimate of 2 prior to seeing the data.

## Model performance

In order to compare the performance of the different models applied, we used the so-called *widely applicable information criterion* (WAIC). This method estimates the pointwise

prediction accuracy of fitted Bayesian models. Here, higher values of WAIC indicate a better out-of-sample predictive fit ("better" model). We refer to Vehtari et al. [24] for more details on WAIC.

### Influence of baseline severity

Because baseline severity has been implied as a potential moderator on the treatment effect of antidepressants, we included it as a predictor in the meta-regression model. We expressed the baseline severity as a fraction of the respective maximum value, in order to account for the occurrence of different depression scales (S1 Table in S1 File). This yielded the model below, where $BL_i$ is the normalized baseline severity of study i and $\gamma$ the slope coefficient for the influence of the baseline severity on outcome variability ratio.

$$lnv_i = \alpha_i + \beta * lnr_i + \gamma * BL_i$$

For this analysis, we excluded studies for which no baseline scores were available and those that used the scale "Hamilton unspecified", leaving a total of 152 studies. The baseline severity scores were normalized according to their respective maximum values.

### Subgroup analysis

In order to investigate the influence of the type of antidepressant of outcome variability, we conducted subgroup analyses applying the REMR model for the following antidepressant classes: SSRI (citalopram, escitalopram, fluoxetine, fluvoxamine, paroxetine and sertraline, 89 trials), SSNRI (desvenlafaxine, duloxetine, milnacipran, venlafaxine, 47 trials), tricyclics (amitriptyline, clomipramine, 11 trials) and atypicals/others (agomelatine, bupropion, mirtazapine, nefazodone, reboxetine, trazodone, vortioxetine, 60 trials).

### Upper bound on the treatment effect heterogeneity

In order to investigate the compatibility of different assumptions regarding the treatment effect with the measured variability ratio, we used the following equation that was derived by A. Volkmann [25]:

$$v^2 - 1 = \frac{(SD_{te} + \rho * SD_p)^2}{SD_P^2} - \rho^2, \text{ where te = individual treatment effect}$$

### Simulation experiments

For each simulation, the response under placebo of 1000 patients was drawn from a right skewed distribution with mean and standard deviation of 8.8 and 7.7 points on the HAMD-17 scale (based on Cipriani data [4]), respectively. For each patient, an individual treatment effect was computed from a mixed Gaussian distribution with a standard deviation $SD_{te}$, where the outcome under placebo and the individual treatment effect were required to be correlated by the correlation coefficient $\rho$. This yielded a potential outcome under placebo and a potential outcome under active treatment for every patient with a corresponding individual treatment effect (see Fig 1 for illustration). Note that only one of these two outcomes can be observed in a real experiment. Half of the patients were then randomly selected for treatment; the other half was assigned to placebo.

For a large treatment effect heterogeneity, the individual treatment effect was drawn from a distribution with an (arbitrarily chosen) standard deviation $SD_{te} = 6.5$ HAMD-17. For a low treatment effect heterogeneity $SD_{te}$ was 1.5 HAMD-17 points.

## Results

### Study selection

We included 169 placebo-controlled studies that reported mean and standard deviation of change in depression scores. These studies included data on 58,687 patients treated with 21 different antidepressants.

### Correlation between mean and standard deviation of depression scores

In order to test the model assumptions of the lnVR and lnCVR meta-analyses, we investigated the linear association between the logarithm of the mean response scores (lnMean) and the logarithm of their standard deviation (lnSD). Fitting a varying intercept model with lnMean as independent variable and lnSD as dependent variable, we got a posterior mean for the slope coefficient of 0.10 with a 95% HDI (highest density interval) interval of [0.04, 0.16]. Fig 4 depicts the linear regression for each depression scale individually. We remark that simply computing the correlation of the two quantities without paying attention to the correct weighting and the different scales in the data would yield an overestimated slope coefficient of 0.25 (see S2 Table in S1 File). These results imply that the lnCVR likely produces biased results, as it assumes a slope coefficient of 1 on this association. The SD will therefore be underestimated in the group with the larger mean (In our case in the antidepressant group). When estimating the direct effect, the lnVR will overestimate the SD in this group, but to a lesser degree, as the estimated slope coefficient of 0.1 is relatively close to 0, which is the slope coefficient assumed by the lnVR.

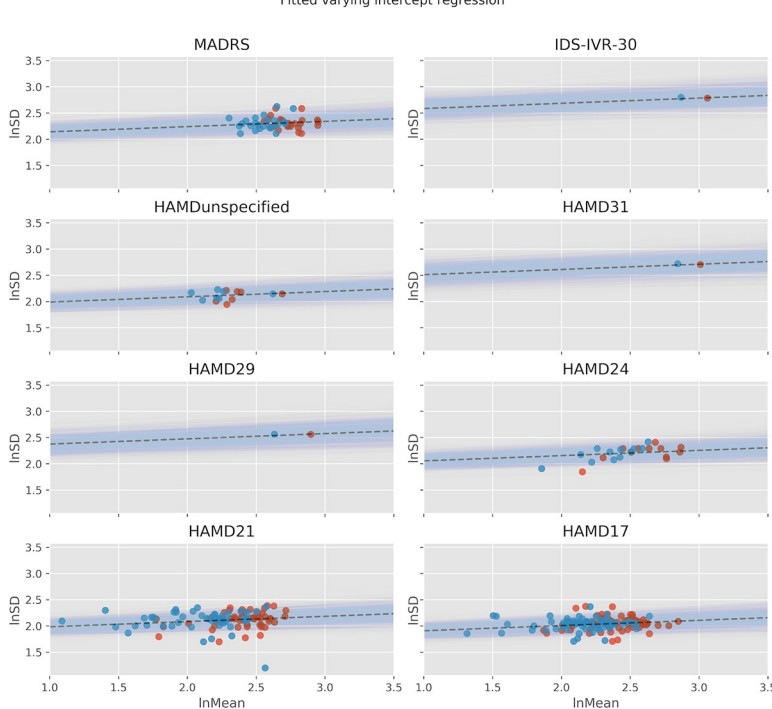

**Fig 4. Linear association between lnMean and lnSD.** Linear association between lnMean and lnSD using a varying intercept model, where the intercepts were allowed to vary between studies with different depression scales. Red dots represent active groups, blue dots represent placebo groups.

## Log variability ratio (lnVR) and log coefficient of variation (lnCVR) models

In order to estimate the difference in variability between antidepressant and placebo response, we modelled the lnVR effect size using a Bayesian random effects model as heterogeneity between studies may be expected. The posterior mean estimate for the variability ratio was 1.01, with the 95% highest posterior density (HDI) interval ranging from 1.00 to 1.02. The lnCVR REMA showed a reduction in the coefficient of variation in the active versus the placebo group (posterior mean estimate for CVR: 0.82, 95% HDI [0.80,0.84]). As predicted, it underestimates the variance in the active group.

## Random-effects meta-regression

Finally, we implemented a Bayesian random effects meta-regression (REMR) that incorporates the strength of the lnMean ~ lnSD relationship directly from the data. The Bayesian shrinkage model is depicted in S2 Fig. Fitting this model, we obtained posterior statistics for the μ and β coefficients (see S3 Table in S1 File and Fig 5). The posterior mean estimate for $e^\mu$ was 1.00 (95% HDI [0.98,1.02]) and that for β 0.04 (95% HDI [-0.03,0.12]), where we can (roughly, up to measurement error and study-specific variation) interpret the coefficients as follows:

$$VR \approx e^\mu * RR^\beta \ (lnVR \approx \mu + \beta * lnRR)$$

Note that the lnVR REMA corresponds to a lnVR REMR with a β coefficient set to 0, whereas the lnCVR REMA corresponds to a lnVR REMR with a β coefficient set to 1. The REMR model learns the β coefficient and its posterior mean is equal to 0.04 [-0.03, 0.12] suggesting that the lnVR REMA is a more appropriate model than the lnCVR REMA when measuring the direct effect on outcome variability. Fig 6 shows the estimates for $e^\mu$ for the different models.

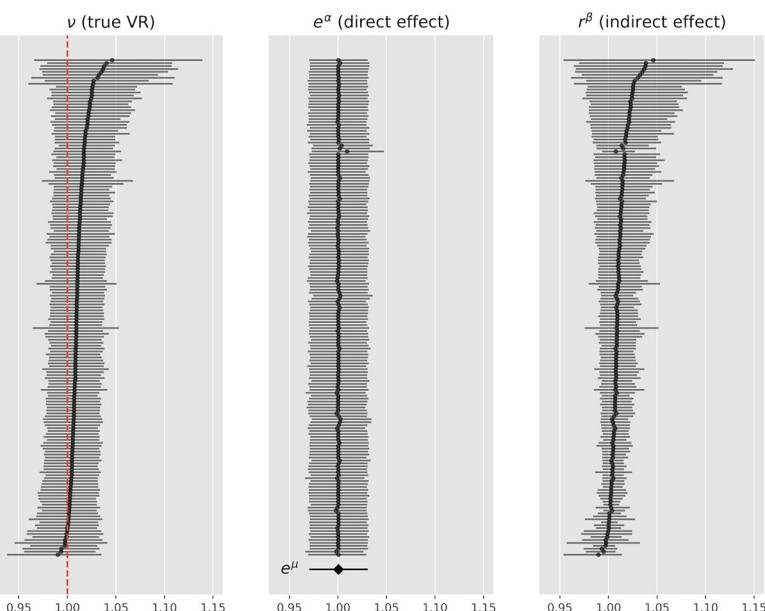

**Fig 5. Posterior credible intervals for total effect, direct effect and indirect effect as determined by the REMR.** $e^\alpha$ represents the study-specific direct effect, $RR^\beta$ the study-specific indirect effect, $e^\mu$ is the meta-analytic mean of the direct effect.

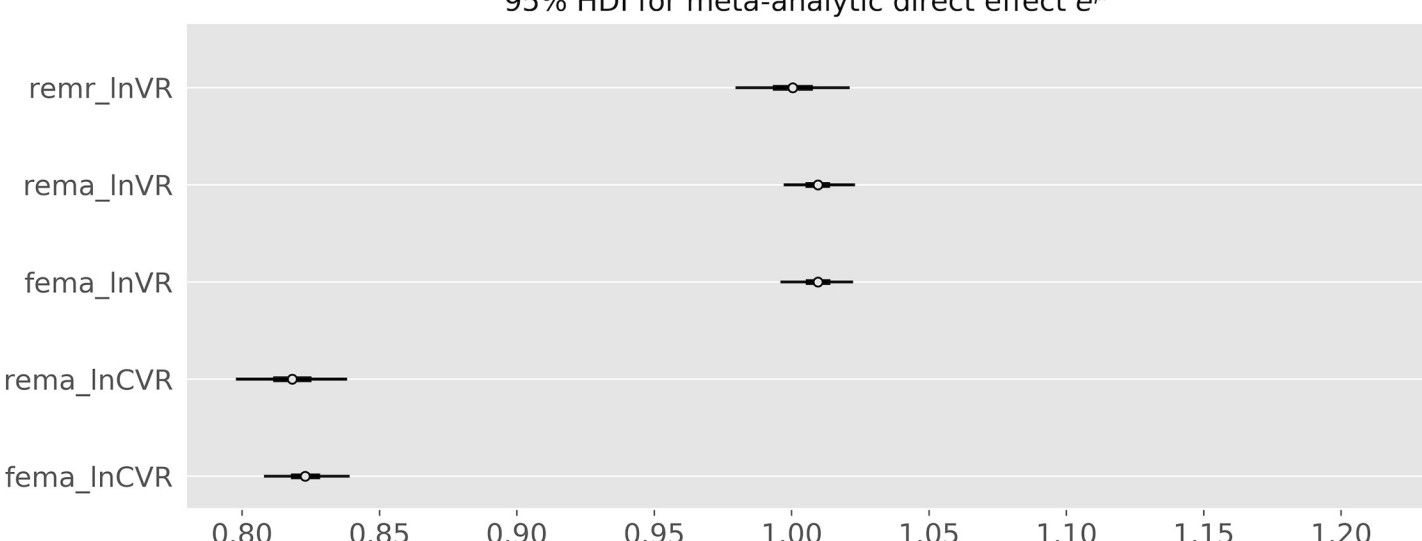

**Fig 6. Posterior credible intervals for the $e^{\mu}$ parameter for the different models.** REMA: random-effects meta-analysis. FEMA: fixed-effects meta-analysis. REMR: random-effects meta-regression. The lnCVR underestimates the variability in the active group, while the results are very similar for the REMR and the lnVR meta-analysis.

We observed that the lnVR REMA and the lnVR REMR outperformed the lnCVR REMA with respect to the WAIC, while the lnVR REMA and the lnVR REMR showed comparable performance. Fig 7 shows the logWAIC for the different models. S4 Table in S1 File depicts the detailed results.

## Between-study heterogeneity

The between-study standard deviation τ under the REMA was found to be low for lnVR (95% HDI [0.00,0.02]). Indeed, applying a fixed effects model instead of the REMA for the purpose of sensitivity analysis yielded similar results for the overall mean estimate of lnVR. Heterogeneity was higher for lnCVR (95% HDI for τ [0.05,0.12]).

## Influence of baseline severity

We included the baseline severity as a predictor in the meta-regression model, in order to estimate its influence on outcome variability (Fig 8). The posterior interval for its regression coefficient γ in the meta-regression was -0.01 (95% HDI [-0.21,0.20], S5 Table in S1 File), suggesting no clear influence of baseline depression severity on outcome variability. The interpretation of the parameter γ is as follows, where BL denotes the baseline severity:

$$VR \approx e^{\mu} * RR^{\beta} * (e^{BL})^{\gamma} (lnVR \approx \mu + \beta * lnRR + \gamma * BL).$$

## Subgroup analysis

In search of treatment effect heterogeneity of individual antidepressant classes, we repeated the main analysis (REMR) for each class. The meta-analytic direct effect was not credibly different from 1 for any of the antidepressant classes (Fig 9). For atypicals, SSRI and SSNRI, it was credibly near 1 (95% HDI [0.96, 1.02], [0.97, 1.05] and [0.96, 1.05], respectively). For tricyclics the mean estimate of the direct effect on variability was 0.91 but with a very wide posterior interval (95% HDI [0.60,1.28]), making this estimate very uncertain.

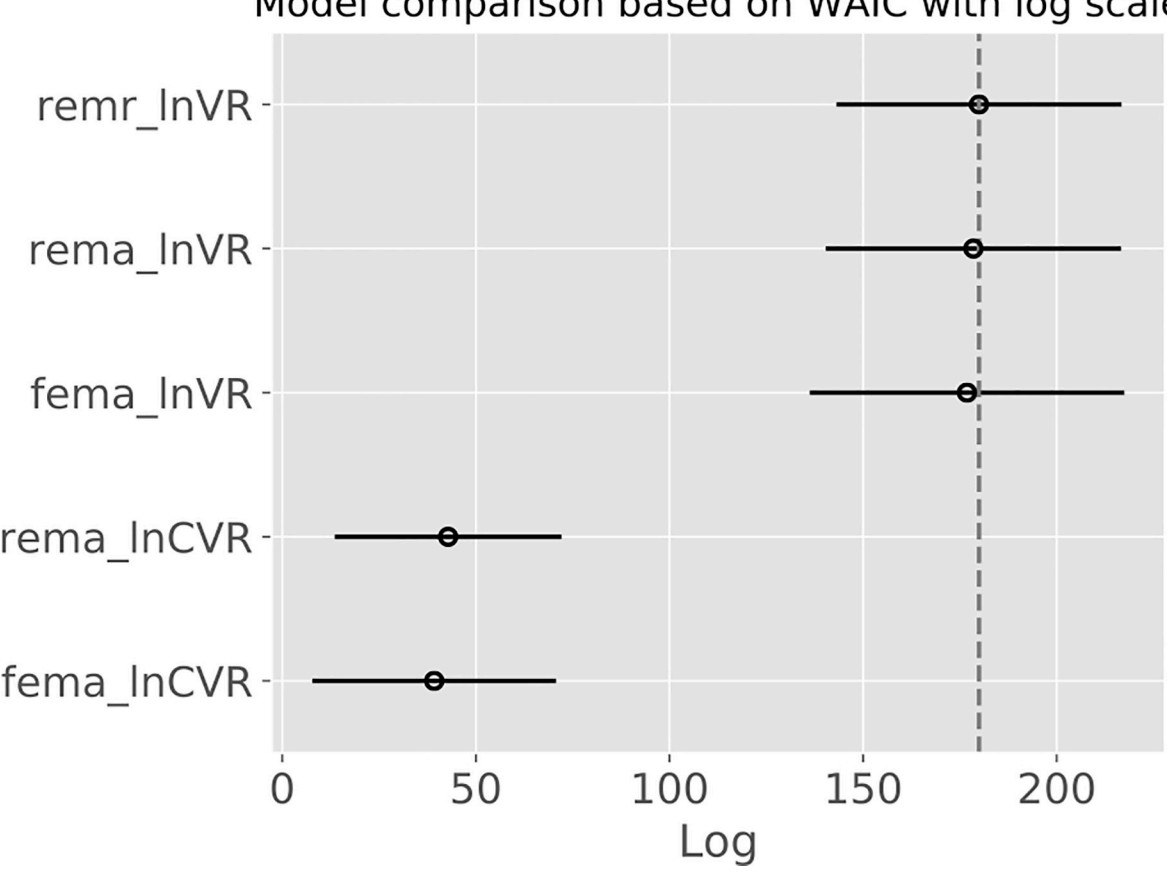

**Fig 7.** *Widely applicable information criterion* **(WAIC) depicted on a logarithmic scale.** Higher values signify a better predictive fit of the underlying model. Bars indicate standard errors. REMA: random-effects meta-analysis. FEMA: fixed-effects meta-analysis. REMR: random-effects meta-regression.

## Sensitivity analysis

Using an enthusiastic prior in the main analysis did not alter the results ($e^\mu$ = 1.00 (95% HDI [0.98,1.02])).

## Upper bound on treatment effect heterogeneity

Using inequation 5 from the methods section, we derived an upper bound on the treatment effect heterogeneity. Using the Cipriani et al. dataset, we can estimate $SD_p$ (SD of placebo response) to be equal to around 7.66 on the HAMD-17 scale. From our main analysis we have that the upper bound of the 95% HDI of the posterior distribution of VR (more accurately $e^\mu$) is 1.02. This implies the following inequality:

4. $SD_{tx} \leq 7.66 * (\sqrt{(0.04 + \rho^2)} - \rho)$

## Which distributions of the treatment effect are possible for a VR 1.02?

Based on formula 6, Table 1 depicts different magnitudes of treatment effect heterogeneity compatible with an average treatment effect of 2.0 HAMD17 points and a VR of 1.02.

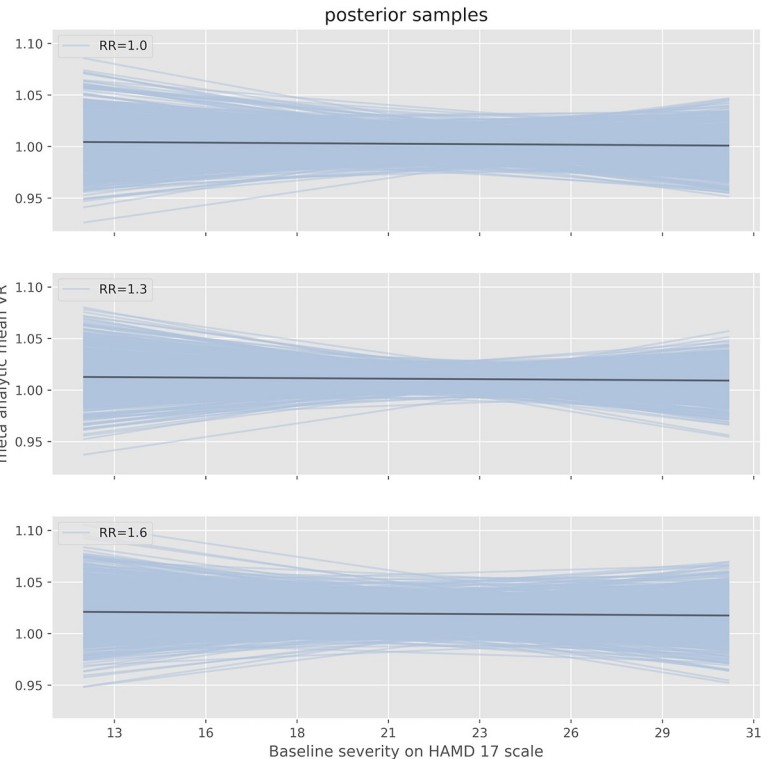

**Fig 8. Influence of baseline severity on outcome variability.** Estimated VR of antidepressants versus placebo as a function of baseline severity on the HAMD17-scale as determined by the REMR. Depicted are the mean estimates and 95% HDIs for three different values of the response ratio (mean response active versus mean response placebo). The meta-analytic mean of RR is around 1.22 in the Cipriani data set.

The left column ("any distribution") of the table depicts the upper bound for the standard deviation of the treatment effect, without any assumptions regarding the shape of its distribution. The mid column shows the respective percentage of patients with a treatment effect

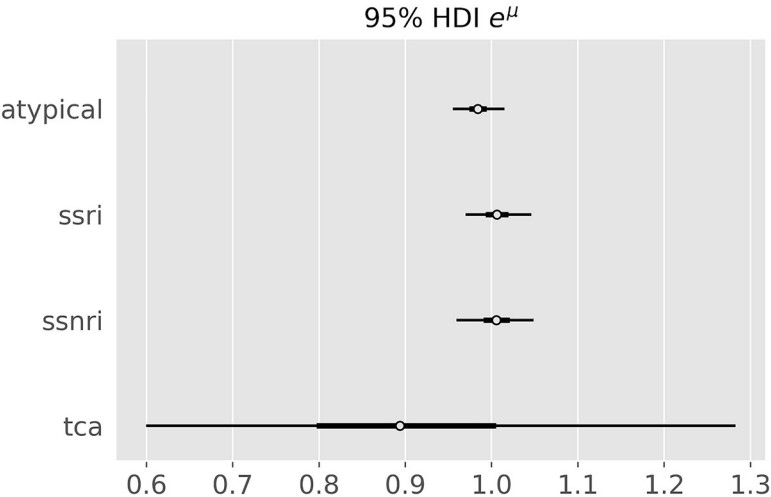

**Fig 9. VR for different antidepressant classes.** Posterior credible intervals for the $e^\mu$ parameter for different antidepressant classes as determined by the random-effect meta-regression.

**Table 1. Upper bound on treatment effect heterogeneity compatible with the results.**

| ρ | Any distribution | Normal distribution | Dichotomous response | |
| | SD$_{te}$ (HAMD17) | % of patients with TE > 7 HAMD17 | Responder TE (HAMD17) | % Responders |
|---|---|---|---|---|
| -1.0 | 15.5 | 37.3 | 9.7 | 20.5 |
| -0.8 | 12.4 | 34.4 | 8.2 | 24.3 |
| -0.6 | 9.4 | 29.8 | 6.7 | 29.8 |
| -0.4 | 6.5 | 22.1 | 5.2 | 38.1 |
| -0.2 | 3.7 | 8.9 | 3.9 | 51.9 |
| 0.0 | 1.5 | 0.1 | 2.8 | 72.2 |
| 0.2 | 0.6 | 0.0 | 2.3 | 86.2 |
| 0.4 | 0.4 | 0.0 | 2.2 | 91.6 |
| 0.6 | 0.3 | 0.0 | 2.1 | 94.1 |
| 0.8 | 0.2 | 0.0 | 2.1 | 95.5 |
| 1.0 | 0.2 | 0.0 | 2.1 | 96.3 |

Assuming a VR of 1.02, a SD$_p$ of 7.66 points (based on Cipriani [10]) and different correlation coefficients ρ between the response under placebo and the treatment effect. Left column ("any distribution"): Upper bounds for the standard deviation of the treatment effect. Mid column: Percentage of patients with TE greater than 7 HAMD17 points, when assuming a normal distribution of the TE with mean 2.0. Right column ("dichotomous response"): Patients are either "non-responders" with a treatment effect of 0, or "responders" with the responder treatment effect. For a given correlation coefficient ρ, there is one possible solution for the treatment effect (TE).

greater than 7 HAMD points when assuming a normally distributed treatment effect. The right column assumes a dichotomous treatment effect ("responder", "non-responder"). Here, "non-responders" are assumed to have a treatment effect of 0 (placebo response = antidepressant response), whereas "responders" have a fixed treatment effect > 0. The percentage of "responders" and their respective "responder treatment effect" depend upon the intra-individual correlation ρ between the potential outcome placebo response and individual treatment effect.

These results show that, contrary to intuition, a variability ratio of 1.02 is (theoretically) compatible with a standard deviation of the treatment effect between 0 and 15.5 points on the HAMD-17 scale. Conversely, a reduction in the variability in the treatment group is compatible with a substantial treatment effect heterogeneity if response under placebo and individual treatment effect are negatively associated (see simulations in S5 and S6 Figs).

## Simulations

We conducted simulation experiments in order to illustrate the compatibility of a VR of 1.02 with different degrees of treatment effect heterogeneity. For a large TEH with SD$_{te}$ = 6.5 HAMD-17, the response under placebo and the individual treatment effect have to be negatively correlated by the correlation factor ρ = - 0.4, in order for the VR to credibly remain at or below 1.02 (see Table 1). Fig 10 depicts the change scores of 1000 patients under placebo (blue) and under active treatment (red) for SD$_{te}$ = 6.5 and ρ = - 0.4. Fig 11 shows the magnitude of the individual treatment effect of 100 individuals of this simulation.

Note that patients that would remain unchanged under placebo (response close to 0) have a larger benefit from active medication (higher density of blue slopes) than patients that would have improved under placebo. For some patients to benefit substantially, however, other patients have to be harmed by the medication (red slopes). Without such a correlation, a TEH of this magnitude would yield a VR of around 1.3.

## Histogram of potential outcome response under placebo and active treatment

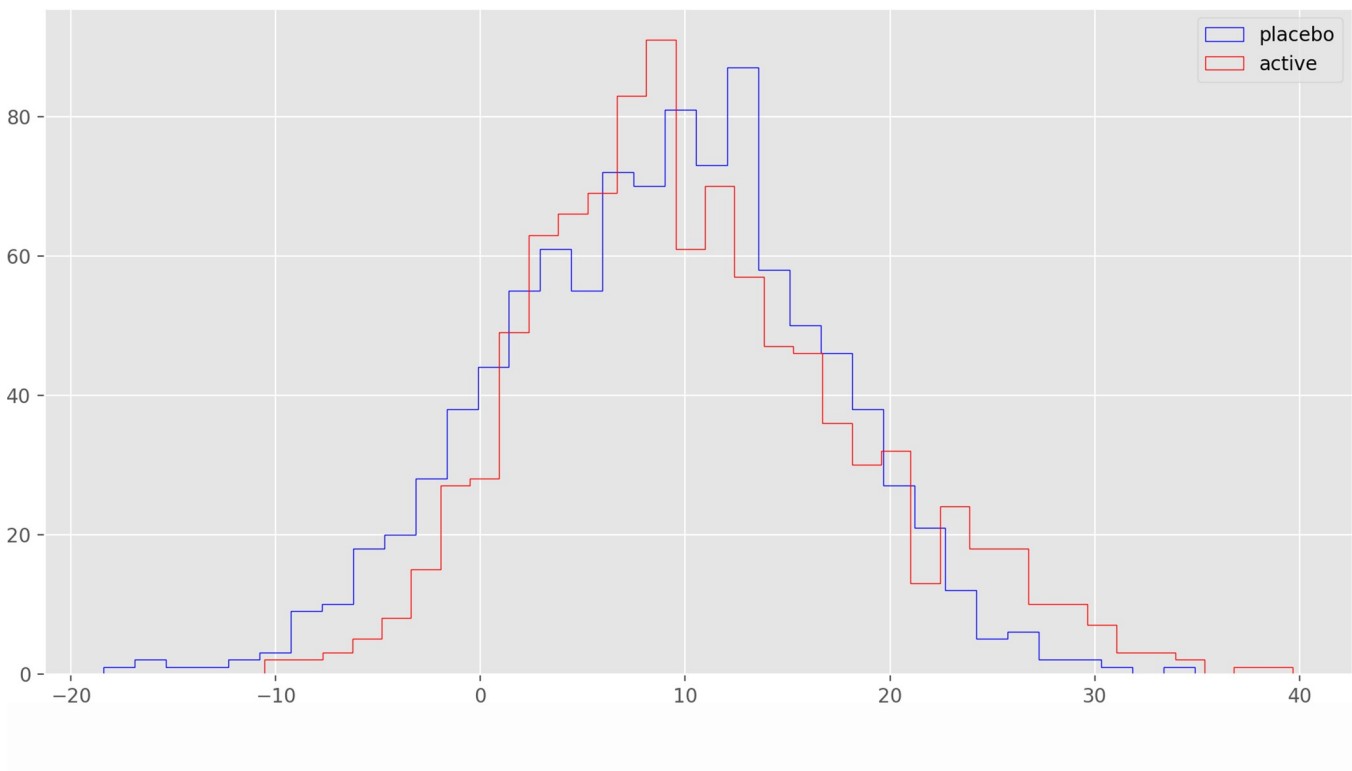

## Potential outcome responses with baseline gauged to 0

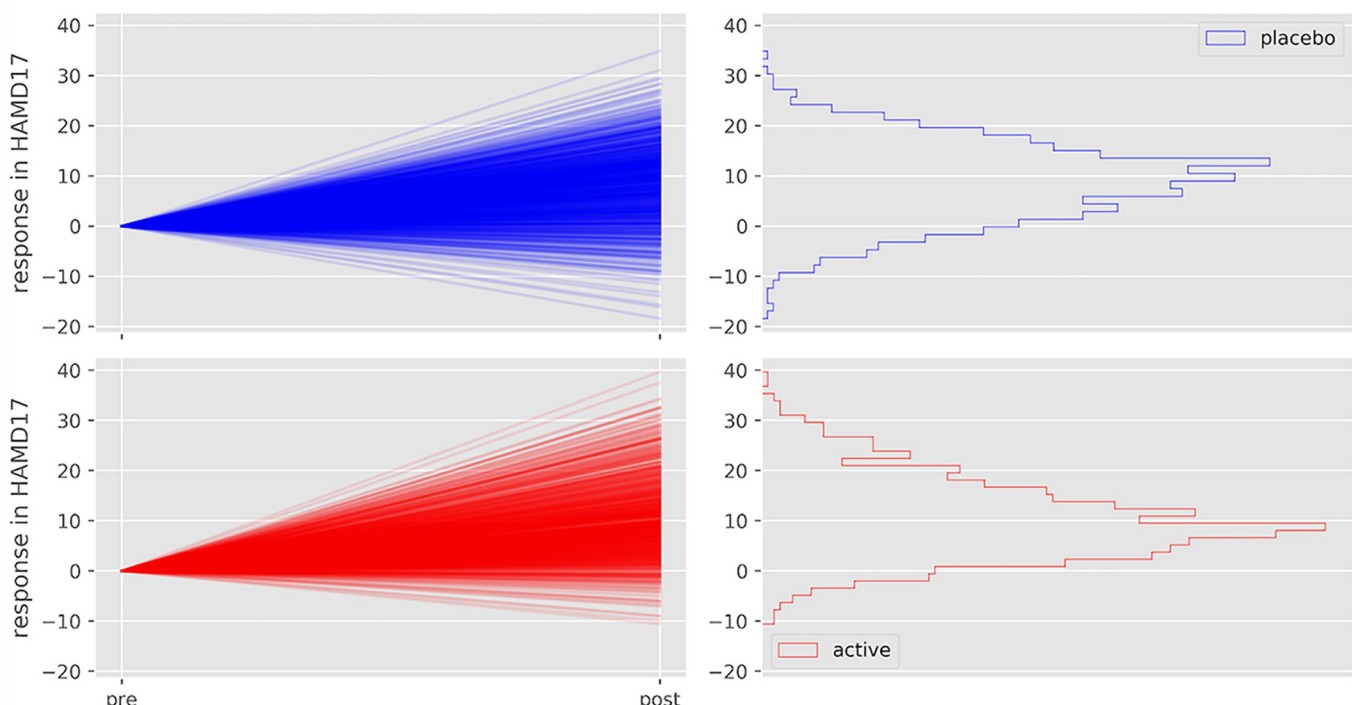

**Fig 10. Change score of 1000 simulated patients.** Boxplot (a) of change score under placebo (blue) and under active treatment (red) for $\rho = -0.4$, $SD_{te} = 6.35$ HAMD-17 points and VR = 1.02. (b) depicts pre-post change for each individual patient, gauged to 0. Note, that in this particular simulation, the $SD_{te}$ is not exactly equal to 6.5, as all simulations contain random processes.

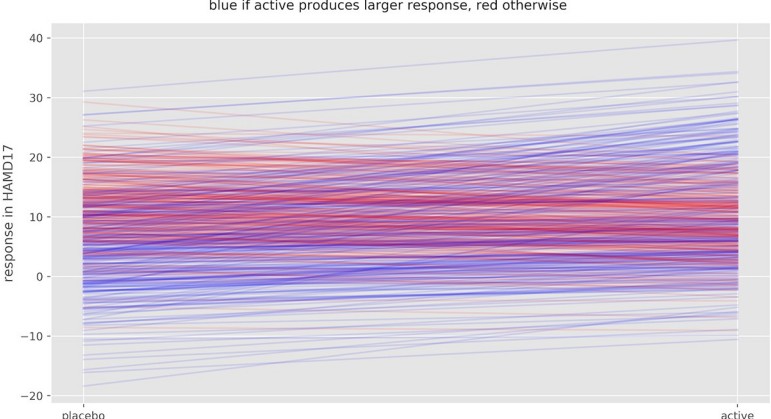

**Fig 11. Potential outcomes and individual treatment effect of 100 simulated patients.** Potential outcome under placebo and under active treatment of 100 simulated patients, $\rho = -0.4$, $SD_{TE} = 6.35$ and VR = 1.2. Slopes represent individual treatment effect, which varies substantially in this simulation. Blue lines indicated improvement under active treatment, red lines deterioration.

For the VR to be credibly lower than 1.02 and the response under placebo and the individual treatment effect to be uncorrelated ($\rho = 0$), TEH has to be low. S3 and S4 Figs depict the results of a simulated experiment with $SD_{te} = 1.5$ points on the HAMD-17 scale (derived from Table 1). A VR closer to 1 would yield an even smaller treatment effect heterogeneity.

## Discussion

Antidepressants are a first-line treatment for patients with major depressive disorder and are widely prescribed around the world. Despite their widespread use, it remains unclear how many and which patients benefit from this medication. In a recent re-analysis [6] of a highly cited network meta-analysis [4], the average treatment effect of antidepressants was found to be about 2 points on the HAMD-17 scale, which is almost undetectable by clinicians [7] and lies below the assumed minimally clinically relevant effect [8, 9]. While this does not exclude the possibility of subgroups of patients that have a meaningful benefit, substantial differences in the individual treatment effect of antidepressants have not been shown to exist yet.

In search for evidence of benefiting subgroups, this work aimed to estimate the treatment effect heterogeneity of antidepressants in the treatment of major depressive disorder using a large dataset of a recent network meta-analysis [4]. To this end, we applied the effect size statistics lnVR and lnCVR suggested by Nakagawa et al. [12], using a Bayesian random-effects meta-analytical approach (REMA) and fitted a multi-level meta-regression (REMR) model to estimate the treatment effect variability between antidepressants and placebo.

Both the lnVR REMR and the lnVR REMA, which were found to outperform the lnCVR REMA, showed that the variability ratio was very close to 1 (REMR: 95% HDI [0.98, 1.02], REMA: 95% HDI [1.00, 1.02]), compatible with a near-constant effect of antidepressants on depression severity.

### Subgroup and sensitivity analyses

We repeated the analyses for different antidepressant classes (SSRI, SSNRI, tricyclics and atypicals) but did not detect clear evidence for treatment effect heterogeneity for any of the groups. The estimates for SSRI, SSNRI and atypicals were relatively certain, while the VR for tricyclics

yielded a wide credible interval with a mean estimate of 0.91 (95% HDI [0.60,1.28]), not clearly suggestive of treatment effect heterogeneity.

Furthermore, we could not detect a relevant influence of baseline severity on outcome variability. Hence, even for severely depressed patients we found no evidence of subgroups that have substantial benefit. This is in line with more recent analyses [26, 27] that could not confirm the previously assumed influence of baseline severity on antidepressant efficacy [28, 29]. In order to test the robustness of our finding, we repeated the main analysis using a more optimistic prior, which did not alter the result.

## Methodological aspects

Other research groups have addressed the same research question [14–16] using the same dataset but have not considered important methodological aspects regarding the lnVR and lnCVR effect sizes which demanded further inquiry.

When estimating the direct effect of antidepressants on outcome variability, the correlation between the mean and standard deviation of the underlying measuring scale has to be taken into account. In other words, how much of the difference in (logarithmic) variances is explained by the difference in (logarithmic) means rather than directly by the treatment. The lnVR and the lnCVR effect sizes naively assume a slope coefficient of 0 and 1 on this association, respectively. Both may thus give biased results, if the true slope coefficient differs from the one assumed.

By applying a varying intercept model taking into account the occurrence of different depression scales we could show that the correlation between (logarithmic) mean and (logarithmic) standard deviation is of a small magnitude (slope coefficient = 0.10), indicating that the lnVR is a more appropriate measure as opposed to the lnCVR in this data set. The superiority of the lnVR over the lnCVR was confirmed by the WAIC. Our work adds accuracy to the existing literature, as we implemented a generalized meta-regression model (REMR) that incorporates the slope coefficient for the correlation between lnMean and lnSD directly from the data. This allowed us to disentangle the direct and indirect effect of antidepressants on outcome variability.

Another important aspect that has not sufficiently been taken into account by previous analyses is the occurrence of different depression scales. Naively regressing over all depression scales may create a spurious correlation that is due to different means and SDs of these scales, rather than a true lnMean ~ lnSD relationship. Indeed, a regression over all depression scales yielded a slope coefficient of 0.25, which is 2.5 x as large as our estimate.

We addressed this issue by applying a contrast-based approach in the REMR. Here, each antidepressant arm is compared to the respective placebo arm of the same study before data pooling, thus resolving the issue of different depression scales across studies.

## Upper bound for the treatment effect heterogeneity

Based on a recently derived inequation [25], we were able to established a direct relationship between VR and treatment effect heterogeneity. We demonstrated that a VR of 1.02 (the upper bound of the 95% HDI interval of the REMR) is theoretically compatible with a standard deviation of the treatment effect between 0 and 15.5 HAMD-17 points. This translates into a maximum of 37% of patients with an individual treatment effect of more than 7 points when assuming a normally distributed treatment effect. Such a large treatment effect heterogeneity would require the treatment effect and the response under placebo of a patient to be strongly and negatively correlated. In absence of such correlation, the treatment effect heterogeneity would be negligibly small ($SD_{te}$ = 1.5 HAMD-17 points).

## How should these results be interpreted?

The VR is a measure that can potentially detect evidence for subgroups that benefit more than average from an intervention. A VR that differs substantially from 1 is evidence of such subgroups, while a VR near 1 is compatible with both a small and a large treatment effect heterogeneity. So how should a VR of 1 (95% HDI [0.98,1.02]) be interpreted? For this, consider the following illustration:

**Hypothesis 1 (H1)**. The treatment effect heterogeneity is close to 0 (e.g. 99% of patients have an individual treatment effect of 1 to 3 HAMD points).

**Hypothesis 2 (H2)**. The treatment effect heterogeneity is greater than in H1.

There are now three possibilities:

1. H1 is true and VR ≈ 1 (very close to 1, e.g. 0.98 to 1.02)

2. H2 is true and VR ≈ 1

3. H2 is true and VR ≠ 1 (not very close to 1)

Our results indicate that VR ≈ 1. We can thus rule out the third possibility. From a Bayesian perspective, the probability of H1 being true increases, while that of H2 being true decreases.

In order for H2 to be true and the VR being close to 1, strong assumptions regarding the correlation between the placebo response and the individual treatment effect of antidepressants are necessary. Specifically, those patients whose depression severity would remain unchanged under placebo would need to have the strongest antidepressant medication effect. In the case of no such correlation, the treatment effect heterogeneity of antidepressants is likely low. However, this correlation cannot be determined using summary statistics of randomized controlled trial, as the inter-individual correlation of placebo-response and individual treatment effect does not equal the correlation of sample mean response and average treatment effect across studies. The direction of association may even be reversed, a phenomenon called Simpson's paradox.

## Limitations

A limitation of this work is that we did not conduct an independent literature search and that we did not use individual patient data which we did not have access to. All results are conditional on the model assumptions. However, prior sensitivity analysis and subgroup analyses did not alter the results. Other limitations were discussed by Munkholm et al. [15]. They include the short duration of trials, as a varying treatment effect may only become apparent after a longer period of time, and psychometrical limitations of the depression scales.

## Conclusion

By applying a multiple level Bayesian regression model and simulations, this work could show that the published data on antidepressants in the treatment of major depression is compatible with a near-constant treatment effect, which is also the simplest explanation for the observed data. Although is not possible to rule out a substantial treatment effect heterogeneity using summary data from RCTs, we could show that a substantial treatment effect heterogeneity is only compatible with the published data under strong assumptions that seem rather unlikely. Until the existence of benefiting subgroups has been demonstrated prospectively, the average treatment effect is the best estimator for the individual treatment effect. Since the average

treatment effect of antidepressants probably falls short of clinical relevance, the current prescribing practice in the treatment of major depression should be critically re-evaluated.

## Supporting information

**S1 Checklist. PRISMA 2009 checklist.**
(DOC)

**S2 Checklist.**
(XLSX)

**S1 Fig.**
(TIFF)

**S2 Fig.**
(TIFF)

**S3 Fig.**
(TIFF)

**S4 Fig.**
(TIFF)

**S5 Fig.**
(TIFF)

**S6 Fig.**
(TIFF)

**S1 Data.**
(XLSX)

**S1 File.**
(DOCX)

## Author Contributions

**Conceptualization:** Constantin Volkmann, Alexander Volkmann, Christian A. Müller.

**Formal analysis:** Constantin Volkmann, Alexander Volkmann.

**Methodology:** Constantin Volkmann, Alexander Volkmann.

**Supervision:** Christian A. Müller.

**Visualization:** Constantin Volkmann, Alexander Volkmann.

**Writing – original draft:** Constantin Volkmann, Christian A. Müller.

**Writing – review & editing:** Constantin Volkmann, Alexander Volkmann, Christian A. Müller.

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
