## [Decision Letter · Decision Letter 0]

14 Jul 2020

PONE-D-20-11332

On the treatment effect heterogeneity of antidepressants in major depression. A Bayesian meta-analysis

PLOS ONE

Dear Dr. Volkmann,

Thank you for submitting your manuscript to PLOS ONE. After careful consideration, we feel that it has merit but does not fully meet PLOS ONE’s publication criteria as it currently stands. Therefore, we invite you to submit a revised version of the manuscript that addresses the points raised during the review process.

Carefully attend to the reviewers' thoughtful and extensive comments.  The both found this work interesting and worthwhile.

We look forward to receiving your revised manuscript.

Kind regards,

Alan D Hutson

Academic Editor

PLOS ONE

:

2. Thank you for inclduing your competing interests statement; "I have read the journal's policy and the authors of this manuscript have the following competing interests: CAM received consulting fees from Silence Therapeutics, outside the submitted work. The other authors declare no competing interest."

Reviewers' comments:

Reviewer's Responses to Questions

**Comments to the Author**

1. Is the manuscript technically sound, and do the data support the conclusions?

Reviewer #1: Yes

Reviewer #2: Yes

2. Has the statistical analysis been performed appropriately and rigorously? 

Reviewer #1: Yes

Reviewer #2: I Don't Know

3. Have the authors made all data underlying the findings in their manuscript fully available?

Reviewer #1: Yes

Reviewer #2: No

4. Is the manuscript presented in an intelligible fashion and written in standard English?

Reviewer #1: Yes

Reviewer #2: Yes

5. Review Comments to the Author

Reviewer #1: This is a very interesting paper. The aim is important and worthwhile investigating (although not that novel - other meta-analyses have quantified heterogeneity in these studies) and the methods used are appropriate and state of the art. I will focus on methods and reporting. The abstract is well-written and balanced. Code is freely available on github, which is great.

1) what is the evidence on the weakly-informed priors used? why these and not others? how sensitive is the key findings (practically zero heterogeneity) to the selection of priors?

2) page 10, the correlation section, I don't quite follow. perhaps restructure a bit? Basically the correlation betwen mean and SD needs to be low and its computed for each effect size type, and the one with the lowest correlation is selected? if that's the case, then readers will get lost a bit. the authors start by stating a comparison is performed and then it is not mentioned again. Or do they mean they just measure the correlation to model it and not introduce bias? or both?

3) Some aspects of the paper are tiring, e.g. explaining relatively aspects of meta-analyses on page 21 (fixed vs random) just provide a reference. You end up with a massive document that seems to be preaching to expert methodologists (Which is fine) but also novice meta-analysts! What is your readership? Can't have both.

4) Unless i've missed it, I'd have expected much more to be done with comparisons of heterogeneity estimates from other studies e.g. the Cipriani Lancet(?) paper. how do they compare, why are they different if they are etc. That should be a whole paragraph in the discussion. that would be much more useful than an explanation of random and fixed effect models. There is a sentence on page 18 referencing 2 studies, but it seems very little considering the whole premise of the paper is this.

Reviewer #2: The authors take on an important task of addressing one of the common assumptions permeating the field by using a large set of clinical trial data to investigate treatment heterogeneity. These analyses are potentially quite important and potentially innovative, but the background and introduction, in particular, could benefit from a more objective vantage that truly appreciates and integrates the breadth of this literature rather than arbitrarily selecting certain references for rhetorical purposes. I happen to agree with the authors that the effects of antidepressants are exaggerated and that compelling evidence for variables that moderate response to identify subgroups more likely to benefit (beyond severity of depression) is lacking. Thus, even though the authors findings and argument does not challenge my scientific viewpoint, there were several areas where the presentation and discussion displayed bias to make some point. This potentially undermines the credibility of what otherwise could be an important work so I’ve tried to point out where possible.

Critical point: References 34 and 35 appear to take on the same aims as this paper and analyze the same dataset. This was not reported until the discussion. This should be made crystal clear in the introduction and background and the authors have to make a case for why a third such analysis is needed to the reader. The value of the paper hinges on the authors ability to do that authentically and any overlap should be acknowledged.

Other points:

1) Introduction, paragraph 1, 1st sentence. “Depression” is not necessarily a disorder. Do the authors intend to focus on major depressive disorder as the title suggests?

2) I do not agree with the comment that antidepressants are broadly debated as an effective treatment. While some may indeed debate this, it seems the broader debate is more about how effective antidepressants are and whether and for what groups this crosses a threshold of clinical significance.

3) The HamD threshold of 3 has also been articulated in the UK NICE guidelines.

4) While I don’t disagree that the 50% improvement threshold is arbitrary, it is a very common a priori defined primary outcome for many trials. This may be worth noting as otherwise it implies the prior meta-analysis picked some arbitrary threshold.

5) Can you really claim as the methods do that this study includes “all RCTs”? Please clarify.

6) Please specify whether or not the authors made any attempt to get individual level data from

7) The authors and editor may want to consider whether the PRISMA diagram should be relegated to a supplementary figure. I think it is critically important and should not be relegated to a supplement. It may be helpful to have more detail about why certain studies were excluded. For how many was it due to not using the specific depression rating scales and for how many for not including continuous data.

8) Could the authors clarify whether they used study measures of effect size and variability from prior reports or if these were recalculated from individual level data for each study?

9) Other meta-analyses have found that antidepressants separate more from placebo the more severe the baseline symptoms. Did the authors look to see if their variability ratio differed by baseline severity of symptoms?

10) Discussion, I agree that 2 points is likely undetectable but “the minimally clinically relevant effect of 7 points” is not well justified. The paper cited makes this distinction based only on a very crude CGI scale, which ironically is almost always dichotomized for analyses, the very thing the authors argue against in their introduction.

11) Discussion. The authors state “These findings are in line with those of two recently published meta-analyses of antidepressants using the same dataset (34, 35).” Rather than leave the reader to look at both of these references, it would behoove the authors to explain what these analyses found and what the authors did that was different. Why was doing another analysis of the same dataset justified? What are these two studies just being cited now and not discussed in the background? (see critical point above)

12) It is very difficult to reviewer the figures with the descriptions buried elsewhere in the text. Even with the description, which I understand, I cannot tell at all what Figure 1 is trying to communicate. Why are some boxes red and some green? Why are they different sizes? How do we determine the treatment effect heterogeneity from a single patient example?

6. PLOS authors have the option to publish the peer review history of their article (what does this mean?). If published, this will include your full peer review and any attached files.

Reviewer #1: No

Reviewer #2: No

---

## [Author Response · Author response to Decision Letter 0]

18 Sep 2020

Dear Reviewers,

First of all, we would like to thank you for your highly valuable input. We appreciate that you made the effort to understand our reasoning and methodology. Before answering your comments point by point, we would like to try to explain what differentiates our paper from previous analyses. 

We believe that other published analyses (Plöderl, Maslej, Munkholm) did not consider important methodological issues when comparing the variability between antidepressants and placebo in the treatment of major depression. 

Crucially, other publications did not take the linear relationship between lnMean and lnSD into account appropriately when estimating the direct effect of antidepressants on outcome variability. The CVR effect size implicitly assumes a slope coefficient of 1 on this relationship. If this assumptions are violated, this effect size yields a biased result. 

The paper by Maslej et al. in JAMA Psychiatry applied the CVR effect size and erroneously concluded that there is evidence for treatment effect heterogeneity. We sent a letter to the journal pointing out the problem with the CVR effect size and the occurrence of different scales (https://www.researchgate.net/publication/342083346_Commentary_on_Maslej_et_al_No_evidence_of_individual_differences_in_response_to_antidepressants).The authors have acknowledged our reasoning and have subsequently retracted their paper from JAMA Psychiatry (https://jamanetwork.com/journals/jamapsychiatry/fullarticle/2767242).

We implemented a meta-regression model that integrates the true slope coefficient directly from the data and is thus more accurate. Furthermore, it allows us to disentangle direct and indirect effect (via the differences in means) of antidepressants on outcome variability. 

All other groups have used a frequentist approach, which merely fails to reject a null hypothesis of equal variance of antidepressants and placebo. In contrast, we applied a Bayesian approach, which allowed us to quantify the range of plausible values for the VR (given the model assumptions). 

Moreover, we demonstrated the exact relationship between the VR and the treatment effect heterogeneity, which has not been done by other publications. 

Reviewer #1: This is a very interesting paper. The aim is important and worthwhile investigating (although not that novel - other meta-analyses have quantified heterogeneity in these studies) and the methods used are appropriate and state of the art. I will focus on methods and reporting. The abstract is well-written and balanced. Code is freely available on github, which is great.

1) what is the evidence on the weakly-informed priors used? why these and not others? how sensitive is the key findings (practically zero heterogeneity) to the selection of priors?

Response:

Thank you for this relevant point. As the amount of data is very large, we used minimally (almost non-) informative, regularizing priors. For example, for the μ parameter in the REMR model we used a Cauchy (0,1) prior. 

We agree that it is appropriate to test the influence of different priors. We have therefore conducted a sensitivity analysis using an optimistic normal prior, centering the mean at 2 (suggesting a 100% increased variance in the antidepressant group) with a 5% probability of VR below 1. This did not alter the results. 

2) page 10, the correlation section, I don't quite follow. perhaps restructure a bit? Basically the correlation betwen mean and SD needs to be low and its computed for each effect size type, and the one with the lowest correlation is selected? if that's the case, then readers will get lost a bit. the authors start by stating a comparison is performed and then it is not mentioned again. Or do they mean they just measure the correlation to model it and not introduce bias? or both?

Response:

Thank you for bringing up this point, which is a crucial point in our analysis, distinguishing it from previous analyses. We have tried to make it clearer in the paper (see page 8). 

We can think of the problem this way: 

If we want to estimate the direct effect of antidepressants on outcome variability, 

If we want to know how much of the difference in variability between two groups (AD and Pbo) is due to the treatment, we need to assess how much of the difference in SDs is explained by the difference in means, such that we don’t falsely attribute it to the treatment variable. If we control for the differences in means, we can get the direct effect of ADs on outcome variability (see new figure 3). 

The CVR effect size controls for the difference in means but makes rigid assumptions on the relationship between lnMean and lnSD, implicitly assuming a slope coefficient of 1 on this relationship. If this assumption is violated, the CVR gives a biased result. 

In the first part of the paper, we regress the lnMean and lnSD in order to test the assumptions of both the VR and the CVR. If the slope coefficient is near 0, the VR would be more appropriate. If the slope coefficient is near 1, the CVR would be the better fit.

Our meta regression model (REMR) directly integrates the slope coefficient into the model and is therefore more appropriate/exact.

To answer your last point. Does the correlation need to be low? No, but the assumed correlation should be similar to the real one. 

The “true” slope coefficient is around 0.1, the CVR assumes a slope of 1 and is thus far off and hence not an appropriate effect size. 

3) Some aspects of the paper are tiring, e.g. explaining relatively aspects of meta-analyses on page 21 (fixed vs random) just provide a reference. You end up with a massive document that seems to be preaching to expert methodologists (Which is fine) but also novice meta-analysts! What is your readership? Can't have both.

Response:

We have tried to shorten parts of the paper that seemed too long/tiring. Specifically, we moved several paragraphs to the methods section (e.g. random vs. fixed effects) and deleted others. 

We agree that we mix highly technical language with a language directed at a readership of clinicians. As we regard the methodology to be the strength of our paper, distinguishing it from previous analyses, we have decided to focus on this. 

4) Unless i've missed it, I'd have expected much more to be done with comparisons of heterogeneity estimates from other studies e.g. the Cipriani Lancet(?) paper. how do they compare, why are they different if they are etc. That should be a whole paragraph in the discussion. that would be much more useful than an explanation of random and fixed effect models. There is a sentence on page 18 referencing 2 studies, but it seems very little considering the whole premise of the paper is this.

Response:

We have added a more detailed discussion of previous analyses and what distinguishes our paper from these. First in the introduction and secondly in the discussion (under Methodological aspects).

As pointed out above, other publications did not consider methodological issues with the VR/CVR effect size and did not take into account the occurrence of different depression scales. Furthermore, we applied a Bayesian approach, which allowed us to quantify the range of plausible values for the VR (as mentioned above).

Reviewer #2: The authors take on an important task of addressing one of the common assumptions permeating the field by using a large set of clinical trial data to investigate treatment heterogeneity. These analyses are potentially quite important and potentially innovative, but the background and introduction, in particular, could benefit from a more objective vantage that truly appreciates and integrates the breadth of this literature rather than arbitrarily selecting certain references for rhetorical purposes. I happen to agree with the authors that the effects of antidepressants are exaggerated and that compelling evidence for variables that moderate response to identify subgroups more likely to benefit (beyond severity of depression) is lacking. Thus, even though the authors findings and argument does not challenge my scientific viewpoint, there were several areas where the presentation and discussion displayed bias to make some point. This potentially undermines the credibility of what otherwise could be an important work so I’ve tried to point out where possible.

Critical point: References 34 and 35 appear to take on the same aims as this paper and analyze the same dataset. This was not reported until the discussion. This should be made crystal clear in the introduction and background and the authors have to make a case for why a third such analysis is needed to the reader. The value of the paper hinges on the authors ability to do that authentically and any overlap should be acknowledged.

Response: 

We have shortened the general discussion on the efficacy of antidepressants and have focused on the methodological aspect of the VR/CVR. We tried to remove remarks from the introduction that could be considered biased or one-sided. We have included a more detailed discussion of previous/simultaneous analyses on this research question in the introduction and in the discussion section (see Methodological aspects). Please also see our introductory remarks that point out what distinguishes our analysis from other works. 

Other points:

1) Introduction, paragraph 1, 1st sentence. “Depression” is not necessarily a disorder. Do the authors intend to focus on major depressive disorder as the title suggests?

Response:

We have replaced the term depression with major depressive disorder/major depression. 

2) I do not agree with the comment that antidepressants are broadly debated as an effective treatment. While some may indeed debate this, it seems the broader debate is more about how effective antidepressants are and whether and for what groups this crosses a threshold of clinical significance.

Response:

We have adjusted the language here and have deleted the sentence referring to said debate. 

3) The HamD threshold of 3 has also been articulated in the UK NICE guidelines.

Response:

We agree with the reviewer that different thresholds have been suggested and have included this one in our discussion of clinical significance. 

4) While I don’t disagree that the 50% improvement threshold is arbitrary, it is a very common a priori defined primary outcome for many trials. This may be worth noting as otherwise it implies the prior meta-analysis picked some arbitrary threshold.

Response:

We have deleted this passage in order to shorten the introduction. 

5) Can you really claim as the methods do that this study includes “all RCTs”? Please clarify.

Response: 

We have deleted the word “all”. The word all was referring to all RCTs identified by the Cipriani group. 

6) Please specify whether or not the authors made any attempt to get individual level data from

Response:

We have not tried to get individual patient data, as we wanted to focus on summary data (mean, SD). 

7) The authors and editor may want to consider whether the PRISMA diagram should be relegated to a supplementary figure. I think it is critically important and should not be relegated to a supplement. It may be helpful to have more detail about why certain studies were excluded. For how many was it due to not using the specific depression rating scales and for how many for not including continuous data.

Response:

We have included all RCTs from the Cipriani dataset with a placebo arm that reported change scores. For the main analyses, we have not excluded any studies based on their depression scale. We have moved the PRISMA flow chart to the main file. 

8) Could the authors clarify whether they used study measures of effect size and variability from prior reports or if these were recalculated from individual level data for each study?

Response:

We have not recalculated these measures from individual patient data, which we had no access to. We have used the study measures of mean and SD that are publicly available here: https://data.mendeley.com/datasets/83rthbp8ys/2

9) Other meta-analyses have found that antidepressants separate more from placebo the more severe the baseline symptoms. Did the authors look to see if their variability ratio differed by baseline severity of symptoms?

Response:

We have included an additional analysis (page 16), adding the baseline severity as a predictor into the meta-regression model. This variable did not seem to have a detectable influence on outcome variability. 

To our knowledge there are 5 analyses investigating the influence of baseline severity on the antidepressant treatment effect. The 3 newest and 2 largest analyses could not confirm this association, which therefore remains uncertain.

Analyses reporting such an association:

Kirsch I, Deacon BJ, Huedo-Medina TB, Scoboria A, Moore TJ, et al. (2008) Initial Severity and Antidepressant Benefits: A Meta-Analysis of Data Submitted to the Food and Drug Administration. PLOS Medicine 5(2): e45.

5,133 patients from 35 trials, reporting an influence of baseline severity on treatment effect. 

Fournier JC, DeRubeis RJ, Hollon SD, et al. Antidepressant drug effects and depression severity: a patient-level meta-analysis. JAMA. 2010;303(1):47-53. doi:10.1001/jama.2009.1943

718 patients from 6 studies. 

Newer studies could not confirm this effect and did not find an influence of baseline severity on treatment effect:

Furukawa, T. A., Maruo, K., Noma, H., Tanaka, S., Imai, H., Shinohara, K., . . . Cipriani, A. (2018). Initial severity of major depression and efficacy of new generation antidepressants: individual participant data meta-analysis. Acta Psychiatr Scand, 137(6), 450-458. doi:10.1111/acps.12886

2464 patients from 4 studies

Rabinowitz, J., Werbeloff, N., Mandel, F. S., Menard, F., Marangell, L., & Kapur, S. (2016). Initial depression severity and response to antidepressants v. placebo: patient-level data analysis from 34 randomised controlled trials. Br J Psychiatry, 209(5), 427-428. doi:10.1192/bjp.bp.115.173906

10.737 patients from 34 studies

Gibbons RD, Hur K, Brown CH, Davis JM, Mann JJ. Benefits from antidepressants: synthesis of 6-week patient-level outcomes from double-blind placebo-controlled randomized trials of fluoxetine and venlafaxine. Arch Gen Psychiatry. 2012;69(6):572-579. doi:10.1001/archgenpsychiatry.2011.2044

8.477 patients from 37 studies

We have also included a subgroup analysis analyzing different antidepressant classes individually. 

10) Discussion, I agree that 2 points is likely undetectable but “the minimally clinically relevant effect of 7 points” is not well justified. The paper cited makes this distinction based only on a very crude CGI scale, which ironically is almost always dichotomized for analyses, the very thing the authors argue against in their introduction.

Response: 

Trying to answer the question of clinical significance inherently involves a dichotomous response, which is indeed an issue difficult to overcome. We agree that the CGI is a discrete measure and therefore also prone to similar problems. It can therefore merely serve as a reference point, not a strict cut-off. It does have 7 levels, which may be slightly better than dichotomizing (yes/no).

We have included 3 points based on the NICE guidelines as another possible threshold for clinical significance, as you suggested above. 

11) Discussion. The authors state “These findings are in line with those of two recently published meta-analyses of antidepressants using the same dataset (34, 35).” Rather than leave the reader to look at both of these references, it would behoove the authors to explain what these analyses found and what the authors did that was different. Why was doing another analysis of the same dataset justified? What are these two studies just being cited now and not discussed in the background? (see critical point above)

Response:

As mentioned in the introductory remarks, we have included a detailed discussion of previous analyses and what distinguishes our paper from them. 

12) It is very difficult to reviewer the figures with the descriptions buried elsewhere in the text. Even with the description, which I understand, I cannot tell at all what Figure 1 is trying to communicate. Why are some boxes red and some green? Why are they different sizes? How do we determine the treatment effect heterogeneity from a single patient example?

Response: 

In figure 1 we have tried to communicate the epistemological problem of determining the true effect of an intervention and the treatment effect heterogeneity. The causal effect of a treatment is the difference between two hypothetical worlds, which are depicted in the figure: A world in which the patient receives the treatment and one in which he doesn’t. 

As only one of both can occur, we can never know the true effect for a single patient. 

We felt that it is important to conceptualize this problem in order to clarify what the treatment effect means (the difference between these worlds, rather than the observed change), as this understanding is crucial for understanding the concept of treatment effect heterogeneity, which signifies how strongly this causal effect varies among different individuals.

We have slightly adapted the description of the figure, hoping that it is now a bit clearer. The boxes represent the remaining depression severity (red) and the improved depression severity (green).

If the editor or reviewers feel that this figure is not helpful or confusing, we could discard it, too. 

6. PLOS authors have the option to publish the peer review history of their article (what does this mean?). If published, this will include your full peer review and any attached files.

Do you want your identity to be public for this peer review? For information about this choice, including consent withdrawal, please see our Privacy Policy.

Reviewer #1: No

Reviewer #2: No

---

## [Decision Letter · Decision Letter 1]

2 Oct 2020

PONE-D-20-11332R1

On the treatment effect heterogeneity of antidepressants in major depression. A Bayesian meta-analysis and simulation study

PLOS ONE

Dear Dr. Volkmann,

Thank you for submitting your manuscript to PLOS ONE. After careful consideration, we feel that it has merit but does not fully meet PLOS ONE’s publication criteria as it currently stands. Therefore, we invite you to submit a revised version of the manuscript that addresses the points raised during the review process.

ACADEMIC EDITOR:There are still two key points from reviewer 2 that need to be addressed.  These appear to be readily fixable. 

We look forward to receiving your revised manuscript.

Kind regards,

Alan D Hutson

Academic Editor

PLOS ONE

Reviewers' comments:

Reviewer's Responses to Questions

**Comments to the Author**

1. If the authors have adequately addressed your comments raised in a previous round of review and you feel that this manuscript is now acceptable for publication, you may indicate that here to bypass the “Comments to the Author” section, enter your conflict of interest statement in the “Confidential to Editor” section, and submit your "Accept" recommendation.

Reviewer #1: All comments have been addressed

Reviewer #2: (No Response)

2. Is the manuscript technically sound, and do the data support the conclusions?

Reviewer #1: Yes

Reviewer #2: Yes

3. Has the statistical analysis been performed appropriately and rigorously? 

Reviewer #1: Yes

Reviewer #2: Yes

4. Have the authors made all data underlying the findings in their manuscript fully available?

Reviewer #1: Yes

Reviewer #2: Yes

5. Is the manuscript presented in an intelligible fashion and written in standard English?

Reviewer #1: Yes

Reviewer #2: No

6. Review Comments to the Author

Reviewer #1: I am happy with the authors' responses and the resulting changes to the document. I hope the process has improved the paper.

Reviewer #2: After reading the response to review, it didn't seem the introduction made clear that prior analyses (and what analyses) used the exact same dataset. This was my most critical point, highlighted as “critical point” in the review. It should be explicitly clear to the reader throughout that two prior analyses used this same dataset and that you are using the same dataset and it should be directly reported how these analyses are different, not indirectly by describing them as if they were studies from separate samples. Be direct and clear about what they did not do and what you are now doing. In addition to this, in the final paragraph of the introduction, you should state that the present study is using the same sample as those prior studies.

Please be sure specify in the manuscript for the reader that you did not make any attempt to get individual level data and that the effect sizes were from not recalculated from individual level data. The limitations do also make this clear.

Figure 1 is now more clear.

7. PLOS authors have the option to publish the peer review history of their article (what does this mean?). If published, this will include your full peer review and any attached files.

Reviewer #1: No

Reviewer #2: No

---

## [Author Response · Author response to Decision Letter 1]

5 Oct 2020

Dear Reviewers, 

thank you for your comments. 

Please consider the last part of the introduction beginning at line 109. We have tried to make it as clear as possible that we reanalyzed a previously analyzed dataset. We have furthermore added a passage explaining the difference of our analysis to the previous ones. See the passage here: 

"Three other research groups (14–16) have addressed the same question of treatment effect heterogeneity of antidepressants using the same dataset.

However, all previous analyses did not sufficiently take into account the linear association between the logarithmic mean and logarithmic standard deviation (SD) of the underlying measuring scale when estimating the direct effect of antidepressants on outcome variability. The VR and the CVR effect sizes naively assume a slope coefficient of 0 and 1 on this association, respectively, and may thus yield biased results if the slope coefficient differs from the assumed one. Crucially, using the VR in this dataset suggests no evidence for treatment effect heterogeneity (14,15), while using the CVR leads to the conclusion of a variable antidepressant treatment effect (16).

We reanalyzed the Cipriani dataset by implementing a Bayesian meta-regression that models the degree of linear association between lnMean and lnSD directly from the data and thus yields more accurate results. We illustrated the exact relationship between VR and CVR and their respective limitations and disentangled direct and indirect effect of antidepressants on outcome variability. We showed that the CVR is a suboptimal measure, as it makes strong assumptions that are not warranted in this dataset. Furthermore, we investigated the compatibility of the results with different degrees of treatment effect heterogeneity using simulations experiments. Using an analytically derived formula, we provided an upper bound on the treatment effect heterogeneity based on the VR estimate." 

We have further added this sentence in the methods section: “We did not attempt to obtain individual patient data but relied on summary data from the openly accessible dataset.”, in order to stress that we used summary data. This is reiterated in the limitations section. 

Best regards, 

Constantin Volkmann

(corresponding author)

---

## [Decision Letter · Decision Letter 2]

16 Oct 2020

On the treatment effect heterogeneity of antidepressants in major depression. A Bayesian meta-analysis and simulation study

PONE-D-20-11332R2

Dear Dr. Volkmann,

We’re pleased to inform you that your manuscript has been judged scientifically suitable for publication and will be formally accepted for publication once it meets all outstanding technical requirements.

Kind regards,

Alan D Hutson

Academic Editor

PLOS ONE

Additional Editor Comments (optional):

Reviewers' comments:

Reviewer's Responses to Questions

**Comments to the Author**

1. If the authors have adequately addressed your comments raised in a previous round of review and you feel that this manuscript is now acceptable for publication, you may indicate that here to bypass the “Comments to the Author” section, enter your conflict of interest statement in the “Confidential to Editor” section, and submit your "Accept" recommendation.

Reviewer #1: All comments have been addressed

Reviewer #2: All comments have been addressed

2. Is the manuscript technically sound, and do the data support the conclusions?

Reviewer #1: Yes

Reviewer #2: Yes

3. Has the statistical analysis been performed appropriately and rigorously? 

Reviewer #1: Yes

Reviewer #2: I Don't Know

4. Have the authors made all data underlying the findings in their manuscript fully available?

Reviewer #1: Yes

Reviewer #2: Yes

5. Is the manuscript presented in an intelligible fashion and written in standard English?

Reviewer #1: Yes

Reviewer #2: Yes

6. Review Comments to the Author

Reviewer #1: I am happy with the authors' responses and the resulting changes to the document. I hope the process has improved the paper.

Reviewer #2: (No Response)

7. PLOS authors have the option to publish the peer review history of their article (what does this mean?). If published, this will include your full peer review and any attached files.

Reviewer #1: No

Reviewer #2: No

---

## [Editor Report · Acceptance letter]

26 Oct 2020

PONE-D-20-11332R2 

On the treatment effect heterogeneity of antidepressants in major depression.
A Bayesian meta-analysis and simulation study 

Dear Dr. Volkmann:

I'm pleased to inform you that your manuscript has been deemed suitable for publication in PLOS ONE. Congratulations! Your manuscript is now with our production department. 

Kind regards, 

on behalf of

Dr. Alan D Hutson 

Academic Editor

PLOS ONE